# A Platform for Ultra-Fast Proton Probing of Matter in Extreme Conditions

**DOI:** 10.3390/s24165254

**Published:** 2024-08-14

**Authors:** Luca Volpe, Teresa Cebriano Ramírez, Carlos Sánchez Sánchez, Alberto Perez, Alessandro Curcio, Diego De Luis, Giancarlo Gatti, Berkhahoum Kebladj, Samia Khetari, Sophia Malko, Jose Antonio Perez-Hernandez, Maria Dolores Rodriguez Frias

**Affiliations:** 1ETSI Aeronáutica y del Espacio, Universidad Politécnica de Madrid, 28040 Madrid, Spain; carlos.sanchez@usal.es; 2Centro de Laseres Pulsados, Building M5, Science Park, Calle Adaja 8, Villamayor, 37185 Salamanca, Spain; tcebriano@clpu.es (T.C.R.); aperez@clpu.es (A.P.); alessandro.curcio@lnf.infn.it (A.C.); ddluis@clpu.es (D.D.L.); ggatti@clpu.es (G.G.); japerez@clpu.es (J.A.P.-H.); frias@clpu.es (M.D.R.F.); 3INFN-LNF, Via Enrico Fermi 40, 00044 Frascati, Rome, Italy; 4Department of Fundamental Physics, University of Salamanca, 37008 Salamanca, Spain; kebladjberkahoum@usal.es (B.K.); samiakhetari@usal.es (S.K.); 5Princeton Plasma Physics Laboratory, 100 Stellarator Road, Princeton, NJ 08536, USA; smalko@pppl.gov; 6Departamento de Física y Matemáticas, University of Alcalá, Plaza de San Diego s/n, 28801 Madrid, Spain

**Keywords:** sensors, sensing technologies, measurements with sensors, ions, ion spectrometers, ion acceleration, magnetic transport, Warm Dense Matter, high repetition rate detection

## Abstract

Recent developments in ultrashort and intense laser systems have enabled the generation of short and brilliant proton sources, which are valuable for studying plasmas under extreme conditions in high-energy-density physics. However, developing sensors for the energy selection, focusing, transport, and detection of these sources remains challenging. This work presents a novel and simple design for an isochronous magnetic selector capable of angular and energy selection of proton sources, significantly reducing temporal spread compared to the current state of the art. The isochronous selector separates the beam based on ion energy, making it a potential component in new energy spectrum sensors for ions. Analytical estimations and Monte Carlo simulations validate the proposed configuration. Due to its low temporal spread, this selector is also useful for studying extreme states of matter, such as proton stopping power in warm dense matter, where short plasma stagnation time (<100 ps) is a critical factor. The proposed selector can also be employed at higher proton energies, achieving final time spreads of a few picoseconds. This has important implications for sensing technologies in the study of coherent energy deposition in biology and medical physics.

## 1. Introduction

The advent of petawatt (PW)-class laser systems in recent decades has paved the way for accelerating ultrashort, brilliant proton bunches. These proton bunches have been utilized to probe matter under extreme conditions, such as Warm Dense Matter (WDM), generated through isochoric heating by lasers, X-rays, electrons, or ions. This area of study is pertinent to High Energy Density (HED) physics [1], laboratory astrophysics, planetology, and particularly laser fusion [2,3,4,5]. HED plasmas, including WDM, represent states of matter that can now be easily generated by lasers or laser-driven secondary sources such as X-rays [6], electrons, or ions [7]. The sub-picosecond dynamics of these states are crucial for understanding the mechanisms of electron–electron and electron–ion thermalization far from equilibrium.

Laser-driven proton sources (LDPSs) are valuable for studying ion stopping power in HED and WDM and for medical applications, including radiography and deflectometry [1,8,9,10,11,12,13]. Ultrafast laser-driven proton bunches are becoming increasingly significant in biology and medical physics, where the effects of rapid dose deposition in tissues are being investigated. Emerging experimental stations now offer combinations of intense solid-state lasers with fourth-generation X-ray free-electron lasers [14], providing highly brilliant photon sources in the keV region. This development enables a new generation of experiments, allowing fast-heated plasma states to be studied using laser-driven electron or proton sources [15] and advanced sensing technologies.

Highly brilliant proton beams are routinely generated via the Target Normal Sheath Acceleration (TNSA) mechanism [16], characterized by an exponential energy spectrum with a sharp energy cutoff reaching up to a few tens of MeV and durations on the order of a few picoseconds. Numerical simulations suggest that improvements in the quality of these sources, in terms of maximum energy and energy spread, can be achieved by increasing laser intensity, enhancing pulse temporal contrast, and improving beam spatial quality [17]. This requires new sensors to validate these simulations. While conventional accelerators can provide proton sources with high brightness, low energy spread, and minimal divergence, LDPSs offer distinct advantages, such as ultrashort pulse duration (∼ ps), high repetition rate capability (up to 1 kHz), and compact size (tabletop machines with dimensions of a few meters and lower cost).

LDPSs are characterized by their extremely short pulse durations, capable of delivering nearly instantaneous doses (less than picoseconds), a feature unmatched by other proton sources [18]. These rapid doses can be comparable to standard doses when operated in high repetition rate (HRR) modes. The compact nature of laser systems compared to conventional accelerators is continually improving. Laser companies now offer compact laser systems at the hundreds of terawatts scale, occupying just a few meters of space. These systems can operate at tens of shots per second, with advancements promising to increase HRR capabilities to 100 shots per second. Smaller systems can achieve tens of terawatts with higher repetition rates, up to 1 kHz, as demonstrated by the Silos system at ELI-ALPS, which operates at 7 fs, 30 mJ, and 1 kHz.

LDPSs are particularly valuable for high-temporal-resolution studies, allowing for the investigation of plasma states on extremely short time scales. By reducing the pulse duration of the “driver” laser to below the hydrodynamic time scale of matter (i.e., below 1 nanosecond), matter can be heated isochorically. This technique, along with the acceleration of short charged particle beams, opens up the possibility for “pump and probe” experiments, where the same laser is used to generate the plasma sample and accelerate the proton probe [19]. However, many such experiments require quasi-monoenergetic and short proton pulses. The TNSA mechanism, while effective, typically produces proton beams with a broad energy spectrum, necessitating additional beam manipulations like energy selection and spatial focusing. Energy selection and beam collimation can be achieved using special configurations of permanent magnets, such as dipoles and quadrupoles. Extensive research has been conducted on focusing and accelerating proton beams using various methods, including helical coil [20,21], varying coil [22,23], hollow micro-sphere [24], micro-lens [25,26], permanent magnet miniature quadropoles [27], proton probing, self-probing [28], and pulsed solenoids [29].

Unfortunately, such manipulations often come at the cost of increased final proton pulse duration, which scales proportionally with the size of the magnetic device used. Several developments in this field are already underway, with notable examples including the ELI-MAIA ion transport system at the ELI Beamlines Facility in the Czech Republic [4,30] and the pair of pulsed solenoids established at the Draco system at HZDR in Dresden, Germany [31]. Additionally, smaller systems have been developed for specific experiments at other laser facilities. [32,33].

These approaches can produce MeV-energy proton beams with time durations in the order of microseconds (μs), or, in the best cases, down to few nanoseconds (ns) mainly due to the distance of proton transport. For instance, a proton beam with a central energy of 1 MeV and an energy spread of 100 keV can experience an increase in temporal length at an approximate rate of 70 ps per centimeter of transport. This presents a challenge for applications requiring precise timing and short pulse durations. The newly opened experimental possibilities involving micrometer-size and picosecond-time plasmas necessitate shorter proton/ion bunches, ideally down to a few picoseconds. For biological sensing applications, it is particularly interesting to achieve even sub-picosecond time durations to study coherent absorption processes.

Recently, an experimental campaign at the Centro de Láseres Pulsados (CLPU) measured proton stopping power in WDM. In this experiment, VEGA -II laser (30 fs 200 TW) was split in two laser beams in order to generate both the WDM sample and the proton beam. The energy loss of the proton beam was measured after passing through the laser-driven WDM sample [13]. This novel experimental approach has enhanced time and energy resolution, paving the way for new experimental activities focused on micrometric-sized and picosecond to sub-picosecond plasmas. These advancements have significant implications in various scientific fields, including inertial confinement fusion (ICF) [2,3] and ion-driven fast ignition scheme of ICF [5,34]. A precise understanding of ion stopping in WDM is crucial for comprehending proton transport in matter [35] and proton isochoric heating [7,36]. These experiments also have applications in studying the structure [37], the equation-of-state [38], and the transport properties of dense plasmas [39], including conductivity [38,40] and thermal equilibrium [41] of WDM samples. Other applications include plasma diagnostics using ion beams [1,11,12] and sensors for HRR solid state detection.

In this experiment at CLPU [13], a compact magnetic selector, combined with an HRR Multi-Channel Plate (MCP) detector [42] was used to manipulate the energy–space properties of the laser-driven proton sources. The proton beam was directed to a laser-generated WDM sample, with a final temporal duration of about 400 ps at the WDM sample, similar to characteristic WDM stagnation time. The central energy of the proton beam was fixed at 500 keV ± 4 keV. The magnetic device was 6 cm long, and the total distance between the proton source and the WDM sample was around 8 cm. The compact magnetic selector consisted of a dipole magnet coupled with a highly sensitive MCP detector, utilizing a system of pinholes and slits at the entrance to control proton divergence, and at the exit to manage the energy band, which could be narrowed at the expense of the total number of selected protons. A new and optimized version of the selector could help reduce temporal stretch; however, achieving durations below 100 ps would require a different approach.

Here, we propose a straightforward method for significantly reducing the duration of proton pulses by using an isochrone ion selector, achieving durations down to a few tens of picoseconds at energies below 1 MeV. These selected proton pulses are ideal for experiments requiring high temporal and spectral resolution, such as ultrashort pump–probe experiments and proton isochoric heating. A particularly relevant application includes proton/ion stopping power studies in warm dense matter (WDM) with unprecedented energy resolution near the Bragg peak region. This is the region where the proton energy is comparable with the thermal energy of WDM electrons, and where many theoretical predictions are currently inconsistent [13]. Additionally, ultrashort LDPSs present new opportunities in biology and medical physics, particularly for studying fast dose deposition in proton therapy. They also have potential applications in material science, such as ultra-short ion implantation.

## 2. Design

In our design, we assume that the proton sources are generated via the TNSA mechanism, which typically produces protons with a characteristic broadband energy spectrum and divergence. The proposed magnetic selector, illustrated in Figure 1, consists of three main components: (i) an initial collimator, (ii) an isochrone magnetic transporter, and (iii) a final energy selector. The collimator comprises two slits (as shown in the lower inset of Figure 1c) placed near the proton source. It selectively allows protons based on their divergence. While this stage introduces some losses, it effectively reduces the angular dispersion of the proton pulse, which in turn minimizes the temporal dispersion at the exit of the magnet. The collimated protons enter into the magnetic transporter made by a dipole (Figure 1a), where a constant magnetic field disperses different proton energies, while maintaining the temporal dispersion. Finally, the energy selector, which includes an exchangeable and movable pinhole (Figure 1b), allows only protons of certain energies to exit. This design enables the production of a highly collimated proton pulse with narrow energy and time spreads. The system offers adjustable parameters: the distance between the slits ($d_{slits}$), the angle between the proton pulse and the side of the magnet (θ), the position ($p_{pinh}$) and radius ($r_{pinh}$) of the pinhole. The first parameter affects the temporal dispersion of the output pulse (ΔT) and is generally minimized. It also influences the angular dispersion at the exit, which is typically very narrow (1–3 mrad). The last three parameters control the central energy ($E_{c}$) and the energy dispersion (ΔE) of the output beam, and they are selected according to experimental requirements.

### 2.1. Temporal Selection

The dynamics of a moving charged particle in a constant magnetic field are governed by the Lorentz force ${\vec{F}}_{L}=q\;\left(\vec{v}\times \vec{B}\right)$, where *q* and $\vec{v}$ are the particle’s charge and velocity, respectively, and $\vec{B}$ is the magnetic field. Since the Lorentz force is always perpendicular to the velocity, it causes particles to follow circular orbits, with the cyclotron radius $R_{c}=mv/qB$, where *m* is the mass of the particle. For non-relativistic protons with energy $E_{c}=mv^{2}/2$ ∼ 2 MeV (i.e., β= u/c ≈ 0.065), the cyclotron radius can be expressed as
(1)$$R_{c}=\frac{\sqrt{2m}}{q}\frac{\sqrt{E_{c}}}{B}.$$

Consequently, in a constant magnetic field, the radius of the proton trajectories depends only on the protons’ energy. If the limits of the dipole are set as shown in Figure 1, the total distance that the proton travels inside the dipole is $D=2\theta R_{c}$, where θ is the angle between the proton’s initial velocity and the magnet side. The total Time of Flight (ToF) of this segment of the trajectory $t_{dipole}$ is
(2)$$t_{dipole}=D/v=\frac{2m}{qB}.\theta$$

Notice that this time does not depend on the energy of the protons but does depend on the angle θ. This implies that the temporal dispersion within the dipole is caused solely by angular dispersion, not by energy dispersion. Therefore, if the entrance angle of the protons is limited to θ±α (as shown in Figure 1c), the temporal dispersion $\Delta t_{\alpha }$ that affects the proton pulse is
(3)$$\Delta t_{\alpha }=\frac{4m}{qB}\alpha .$$

To achieve minimal temporal dispersion, a small acceptance angle is critical, which is why a slit system before the dipole is necessary. However, this system requires protons to travel some distance in free space, leading to increased temporal dispersion due to variations in kinetic energy. The distance between the source and the first slit ($d_{source}$) should be as short as practically possible, while the distance between the slits ($d_{slits}$) is determined by the desired acceptance angle:(4)$$d_{slits}=\frac{h}{\tan(\alpha )}.$$
where *h* is the height of the slits and α is the acceptance angle. The temporal dispersion caused by this initial flight ($\Delta t_{\mathrm{ff}}$), assuming a very small angle α, is
(5)$$\begin{array}{cc} \Delta t_{\mathrm{ff}}= & \left(d_{slits}\left(\alpha \right)+d_{source}\right)\sqrt{\frac{m}{2}}\times \\ \times & \left[\frac{1}{\sqrt{E_{c}-\Delta E})}-\frac{1}{\sqrt{E_{c}+\Delta E})}\right]. \end{array}$$

Those two effects—time dispersion due to angular dispersion ($\Delta t_{\alpha }$) and due to initial flight ($\Delta t_{\mathrm{ff}}$)—have opposite dependencies on the acceptance angle. For a given central energy and energy dispersion, changing the acceptance angle α by adjusting the distance between slits $d_{slits}$, affects these dispersions differently. Increasing the acceptance angle reduces the temporal dispersion of the initial flight but increases the dispersion due to angular effects, and vice versa. The total temporal dispersion, accounting for both effects, is given by
(6)$$\Delta t\left(\alpha ,E_{c},\Delta E\right)=\sqrt{\Delta t_{\mathrm{ff}}^{2}+\Delta t_{\alpha }^{2}}.$$

### 2.2. Energy Selection

To control the spectrum of the exiting protons ($E_{c}\pm \Delta E$), a final pinhole is placed. The expressions of the pinhole’s position ($p_{pinh}$) and radius ($r_{pinh}$) are
(7)$$\begin{array}{cc} p_{pinh} & =k_{E}\;\sin\left(\theta \right)\left(\sqrt{E_{c}+\Delta E}+\sqrt{E_{c}-\Delta E}\right), \end{array}$$
(8)$$\begin{array}{cc} r_{pinh} & =k_{E}\;\sin\left(\theta \right)\left(\sqrt{E_{c}+\Delta E}-\sqrt{E_{c}-\Delta E}\right). \end{array}$$
where $k_{E}=\sqrt{2m}/\left(qB\right)$. It is also useful to express these equations in terms of the central energy and energy dispersion as functions of the pinhole’s position and radius, so they can be checked during the experiment.
(9)$$E_{c}=\frac{\left(p_{pinh}^{2}+r_{pinh}^{2}\right)}{4k_{E}^{2}\sin^{2}\left(\theta \right)},\;\Delta E=\frac{p_{pinh}\cdot r_{pinh}}{2k_{E}^{2}\sin^{2}\left(\theta \right)}.$$

Another useful expression is the maximum proton energy that a dipole can operate as a function of the length of the dipole *L* and the angle θ:$$E_{MAX}=\frac{L^{2}}{4k_{E}^{2}\sin^{2}\left(\theta \right)},$$
(10)$$E_{MAX}\left[MeV\right]\approx \frac{1.2\times 10^{-5}}{\sin^{2}\left(\theta \right)}\;B\left[T\right]\;L^{2}\left[mm\right].$$

Considering a typical case, a dipole of L = 20 cm, *B* = 1 T, and θ = 30^∘^, can select a maximum energy of 1.9 MeV. In theory, any energy is accessible if the angle θ is small enough, regardless of the dipole’s length. However, a very small angle makes the dipole more challenging to align, so a compromise is recommended. Assuming a fixed central energy and energy dispersion, the acceptance angle that achieves the smallest temporal dispersion can be numerically calculated. This angle typically lies between 0.5 and 2 mrad. The total temporal dispersion achieved with these configurations is illustrated in Figure 2. The time dispersion decreases as the ratio $\Delta E/E_{c}$ decreases (i.e., for higher energies and smaller bandwidths). To better evaluate the selector’s performance, we can compare the time spread of a proton beam with the same parameters as in [42]. For a proton energy of 500 keV ± 4 keV, and a distance between the source and the first slit $d_{source}$ of 3 mm, the proton pulse at the exit of the selector can have a final duration of approximately 70 ps. This is about six times smaller than the 400 ps duration obtained with the previous selector in similar configurations. The graph in Figure 2 shows that, for a central energy of 500 keV, even shorter proton pulses (around 50 ps) can be achieved by reducing the energy bandwidth to 15 keV. This can be achieved by either reducing the number of selected particles or by decreasing the proton source size.

An alternative operational regime, featuring sub-picosecond proton pulses (≤1 ps), can be achieved by considering higher proton energies of 15 MeV with a few keVs of energy spread. This will open new possibilities for fast proton imaging and/or deflectometry using high-resolution 2D ion spectrometers with high repetition rates (HRR). According to Equation (10), to achieve 15 MeV of selected energy, the selector configuration must be adjusted by increasing the total size *L* up to 300 mm and reducing the angle θ to $15^{\circ }$.

### 2.3. Transmission Coefficient

The proposed ion beam selector operates through three distinct, consecutive steps:The first collimator selects a quasi-pencil beam from the initial laser-driven protons. The proton source consists of a circular surface with a radius of a few hundred micrometers, which is significantly larger than the slit apertures, typically 10–20 μm in diameter. Spatial selection can be achieved using either a single or double pinhole, depending on the spatial properties of the proton beam. If the proton beam propagates laminar (i.e., individual proton trajectories do not cross), a single pinhole is sufficient. Conversely, if the proton source is randomly distributed, a double-pinhole system is necessary.The second phase involves transporting the selected beam isochronally into the magnetic dipole. During this phase, the energy–time distribution is transformed into an energy–space distribution along the side of the dipole. For higher central energies, the path through the dipole is longer, resulting in a larger final band at the exit of the selector, as measured by the MCP detector.The third and final phase involves selecting the energy band out of the selector. This is carried out using a second movable pinhole along the side of the dipole where the proton energy is spatially distributed. The spatial distribution of the proton source along the side is explained by the equations from (8) to (10).

The three phases contribute to building the final selected beam, but at the cost of a substantial reduction in the total number of particles, which can be described by the transmission coefficient. Any improvements in the total number of protons must be achieved through enhancements in the acceleration mechanisms mentioned earlier in the text, by reducing the initial divergence, which is the primary source of flux reduction.

Based on the description above, the transmission coefficient can be defined as the product of two terms: $\eta_{in}$, which describes particle losses in the first collimator (comprising a series of pinholes at the entrance of the selector), and $\eta_{out}$, which describes particle losses in the energy selector due to the use of a movable pinhole at the exit of the magnetic field. The combination of the entrance and exit pinholes results in a narrow solid angle of acceptance for protons with a specific spatial divergence and central energy band.

We implemented a tracking code to better estimate the transmission coefficient. The calculated transmission coefficient is based on the geometrical scheme proposed in Figure 3 and the result is represented as a function of energy dispersion and different central energies in Figure 4.

A simpler approach for a rough estimation of the transmission can be used. Let us assume proton laminar source generated in a circular area of radius D. Placing a pinhole with radius *r* at a distance *h* will give a transmission factor $\eta_{in}$ which is defined by the ratio of two solid angles $d\Omega_{ph}$ defined by the used pin-hole and the $d\Omega_{0}$ defined by the initial source divergence. Finally,
(11)$$\begin{array}{ccc} \eta_{in} & = & \frac{d\Omega_{ph}}{d\Omega_{0}}=\frac{1-{\left(1+\tan^{2}\alpha_{in}\right)}^{-1/2}}{1-{\left(1+\tan^{2}\alpha_{0}\right)}^{-1/2}},\\ \tan\alpha_{in} & = & \frac{\tan\alpha_{0}r}{D+\tan\alpha_{0}h}. \end{array}$$

Assuming a proton source with a circular surface radius of *r* = 75 μm, a divergence angle of 20^∘^ and a collimator with an 8 μm radius pinhole placed at a distance h=2 mm, we obtain $\eta_{in}=1.1\times 10^{-4}$ (see Equation (12), which is consistent with the estimations shown in Figure 4). Note that the figure shows the total transmission, considering energy dependence and the final energy selector.

If the proton source is not laminar but uniformly distributed within the maximum divergence angle (assumed here as 20 degrees, see Figure 3b), a double-pinhole system must be used. Monte Carlo simulations were performed under similar conditions as those in Figure 3 (i.e., 7.6 mm thick double pinhole with a diameter of 16 μm, and a circular proton source with a 75 μm radius placed 2 mm distance form the pinhole). The two-pinhole system was modeled as two single rectangular cells (see Figure 3a)) with zero importance imposed, meaning particles entering them are excluded, giving us the transmission coefficient (normalized to unity). The result of applying tally F1 (current integrated over a surface) at the exit of the slit yields a transmission factor $\eta_{in}=6\times 10^{-4}$, which is one order of magnitude higher than for the laminar source.

It is important to highlight that such an “apparently low transmission” does not affect the measurement procedure. The authors have successfully demonstrated the feasibility of using and measuring proton beams with particle numbers corresponding to a transmission coefficient on the order of 10^−5^–10^−4^ using dedicated spectrometers and high-resolution and sensitive MCP detectors, as extensively described in [13,42].

## 3. Numerical Simulations

### 3.1. Particle Tracking

In this section, we analyze the general case where the proton source is generated by a circular surface with isotropic directions within a maximum angle of 20 degrees. The conceptual design is executed by tracking individual protons along their path through the device, utilizing the Lorentz force and the Runge–Kutta method implemented in Python with the libraries Numpy, Scipy, Numba, and Matplotlib.

In this scenario, the collimator consists of a double slit system as shown in Figure 1. The simulation parameters are chosen to minimize the temporal spread as described in the previous section and are defined as follows: The distance between the source and the first slit is $d_{source}$ = 2 mm, the distance between slits is 7.6 mm, the slit width is 8 μm. The dipole has a magnetic field of B = 1 T, the angle between the dipole side and the proton beam θ has been set at 60^∘^ (30^∘^ from the vertical axis), and the dipole has a length of 15 cm. The pinhole is placed at a distance $p_{pinh}$ = 10.17 cm from the entrance of the proton beam, and has a radius of 2.3 mm, aimed to the energy range of 0.5 MeV ± 25 keV. The input proton beam has an energy distribution with a maximum cut-off of 4 MeV with a maximum proton energy around 500 KeV. The proton beam source size is simulated as a circular source with a radius of 75 μm and a half angle angular distribution of 20^∘^.

The simulation results are illustrated in Figure 5, which shows the trajectory of the simulated proton beam (left), and compares the energy dispersion and temporal dispersion at the source and at the end (right upper and right lower, respectively). The results indicate that extremely monochromatic and narrow pulses can be obtained at the device’s exit.

These simulations assume a uniform magnetic field, which would correspond to a sector magnet (or a more complex design to minimize the fringe field) in reality. In our simplified design (shown in Figure 6 and Figure 7), a non-ideal interaction length $L_{eff}$ might slightly affect the selector’s performance. However, the necessary tools and knowledge for accurate calibration of the real instrument will be available upon its realization. Specifically, the fringe field formation at the border of the dipole may result in a longer interaction length according to the formula
(12)$$L_{eff}=\frac{\int Bds}{B_{max}}.$$
where *s* follows the mean magnetic path of the particles and $B_{max}$ is the field value in the uniform region. In other words, the average field experienced by the particle will be
(13)$$B_{av}=\frac{L_{no-fringe}}{L_{eff}}B_{max}<B_{max}.$$
Here, $L_{no-fringe}$ is the length of the particle’s path in an ideally uniform field without fringe effects. This effect varies among different magnets, even for optimized designs, and must be considered for the final calibration of the instrument.

Figure 6 and Figure 7 show the numerical simulations using the Finite Element Method software FEMM 4.2 [43] for designing the ferromagnetic structure to optimize border effects. Figure 6 depicts the simulation of the ferromagnetic structure, which combines Pure Iron and Neodymium magnets (N52 grade, 100×200×20 mm, separated by 5 mm). Figure 7 shows the line-out of the magnetic field generated by the dipole. Point A and Point B mark the beginning and end of the magnet, respectively, which has a field of 0.5 T. The field outside the gap at 5 mm is 100 mT, and at 10 mm it is 50 mT. It takes approximately 6 mm to increase from 0.5 T to 1 T and 10 mm to rise from 0.5 T to 1.25 T. The magnet is composed of Pure Iron and Neodymium magnets N52 grade of 100×200×20 mm, separated by 5 mm.

### 3.2. Monte Carlo Simulations

In laser–plasma experiments, protons are generated through laser–solid interactions. After generation, the protons undergo space charge effects, which are crucial for understanding their spatial–time–energy configuration at early stages. These effects are significant in the first hundred femtoseconds and diminish rapidly after a few tens of picoseconds. Beyond this early period, collective effects become negligible, justifying the use of a single-particle approach for simulating proton transport.

Monte Carlo codes are effective tools for designing the different stages of the experiment. Specifically, these codes can simulate an initial wide-spectrum and divergent proton source through a spatial filter, which may consist of a single (laminar beam) or a double (uncorrelated beam) pinhole. The selected beam is then transported through a magnetic dipole, followed by a final pinhole for energy selection. The resulting proton beam can be used to study material properties by analyzing differences in particle stopping power, particle-induced X-ray emission (PIXE), and multiple scattering. Monte Carlo methods can predict and measure these effects. Additionally, the secondary electrons produced by the proton beam’s interaction with the material of the selector can also be simulated and estimated using Monte Carlo simulations.

At this stage of the design, we present a preliminary simulation with parameters similar to those used in the previous sections. Simulations were performed using the MCNP6 software, which includes routines for tracking charged particles through magnetic fields via direct integration methods [44].

The simulation parameters for MCNP6 are as follows: The dimensions of the rectangular cell forming the dipole are 15 cm in length (*Z* dimension), 5 cm in width (*X* dimension), and 0.6 cm in height (*Y* dimension). To save computational time, we reduced the dimensions of the magnet in some cases. The magnet is tilted 30 degrees with respect to the vertical axis (clockwise). The proton source energy is set within the range of 500 ± 25 keV (see Figure 8 inset for the initial energy distribution). The initial beam is modeled as a Dirac delta without width and with no divergence.

The proton source is positioned 4.5 mm from the closest point of the magnet (a distance irrelevant without divergence), and the magnetic field modulus is set to 1 T, with the field direction normal to the magnet surface. Trajectories are analyzed using the FMESH superimposed MESH tally B type FLUX [45] to solve the problem over the system geometry.

The results are shown in Figure 9a, which depicts the proton trajectories as a function of energy. Lower-energy protons exit at 9.95 cm from the input point (0.4 cm from the magnet’s bottom corner in the simulation), while higher-energy protons exit at 10.50 cm. These values align with expression (8), resulting in $p_{pinh}=$ 10.22 cm, which is approximately the midpoint between the two energy values mentioned. We extended the study by adding a small divergence of 0.2 degrees and placing the source 1 mm from the magnet (with the same energy interval). Figure 9b shows the projection of the flux over the magnet surface, while Figure 9c,d show detailed tracking at the magnet exit and the proton spatial profile, respectively. The profile confirms that the calculated pinhole radius is 0.26 cm.

Another outcome of the MCNP6 simulations is the spot shape at the magnet’s exit (with no divergence), where the particle distribution at different energies along the z-axis is shown in Figure 10a. Figure 10b,c display the normalized projections of the spot over the axes of Figure 10a (profiles). These simulations reveal the proton beam’s extension in the Z dimension while remaining sharp in the Y dimension. Using expression (8) with the MCNP6 parameters, the pinhole radius ($r_{pinh}$) is 0.26 cm, which corresponds to half the FWHM in Figure 10c (spot profile projection in Z), confirming the analytical predictions.

## 4. Discussion and Conclusions

A novel and simple design for an isochrone magnetic selector has been proposed, capable of angular and energy selection for laser-driven proton pulses with moderated energies (<2 MeV). This design minimizes temporal duration and facilitates the subsequent measurement of the energy spectrum using high-resolution sensors (HRR). The proposed device can select and measure proton beams with central energies in the hundreds of keV range and bandwidths of a few tens of keV, while maintaining a temporal duration below 100 ps. This is essential for probing micrometric-sized laser-driven warm dense matter (WDM) samples.

A highly sensitive detection system for proton spectroscopy, working at high resolution, has been developed. A preliminary parametric study compares the performance of this new selector with a previous magnetic selector used in recent experiments. For example, 400 ps proton pulses from the experiment referenced in [13] can be replaced by an alternative proton beam with a duration of less than 100 ps (down to 70 ps). Numerical simulations, including particle tracking and Monte Carlo code MCNP6, support these analytical estimations. MCNP6 will also enable integrated simulations during the experiment, including stopping power calculations.

Additionally, the magnetic selector can be adapted for higher proton energies up to 15 MeV by increasing the angle with respect to the proton propagation, thereby significantly reducing the proton pulse duration to the picosecond and sub-picosecond ranges.

## Figures and Tables

**Figure 1 sensors-24-05254-f001:**
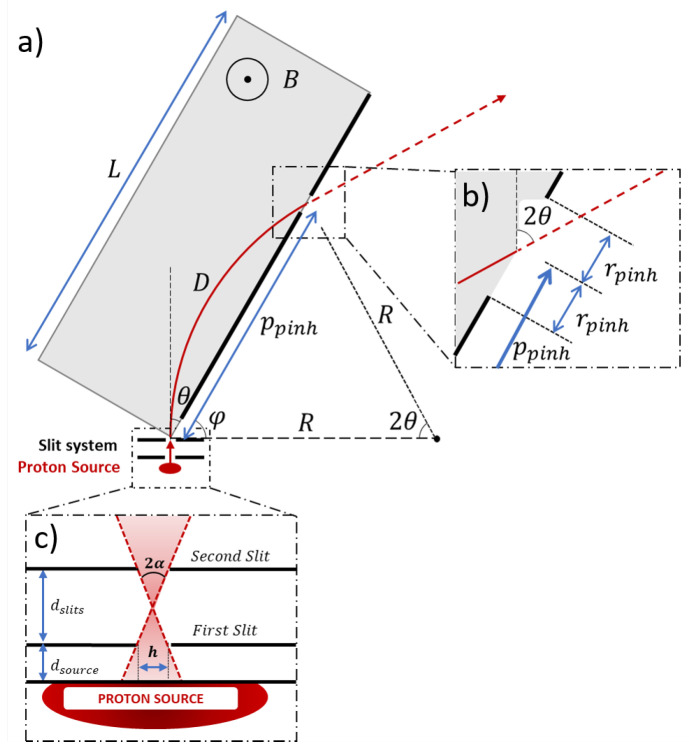
Diagram of the isochrone ion selector. (**a**) Scheme of magnetic transporter. Zoom of the final pinhole and (**b**) the slit system (**c**). Angles θ and φ are complementary.

**Figure 2 sensors-24-05254-f002:**
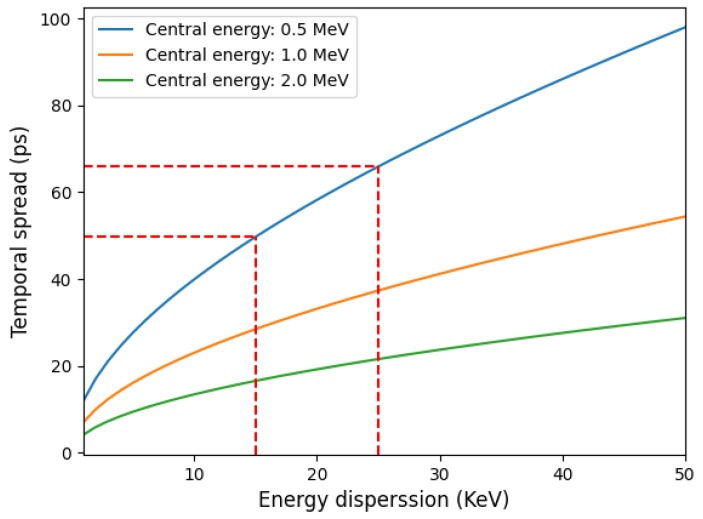
Time dispersion as a function of the energy dispersion for different central energies.

**Figure 3 sensors-24-05254-f003:**
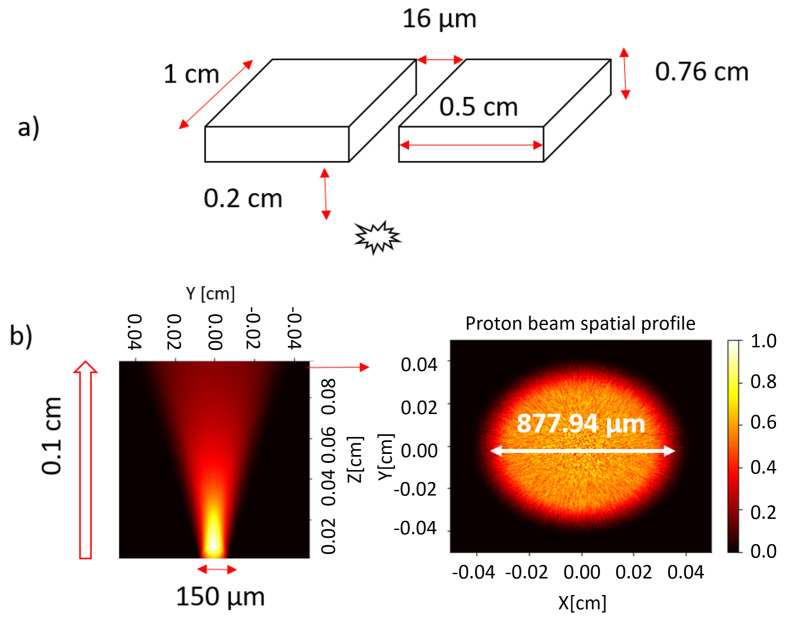
Schematic of the Monte Carlo simulation of the entrance slit: (**a**) geometry of the slit, (**b**) characterization of the source placed 1 mm from the slit with a 20 degrees half-angle divergence and an initial spot diameter of 150 μm.

**Figure 4 sensors-24-05254-f004:**
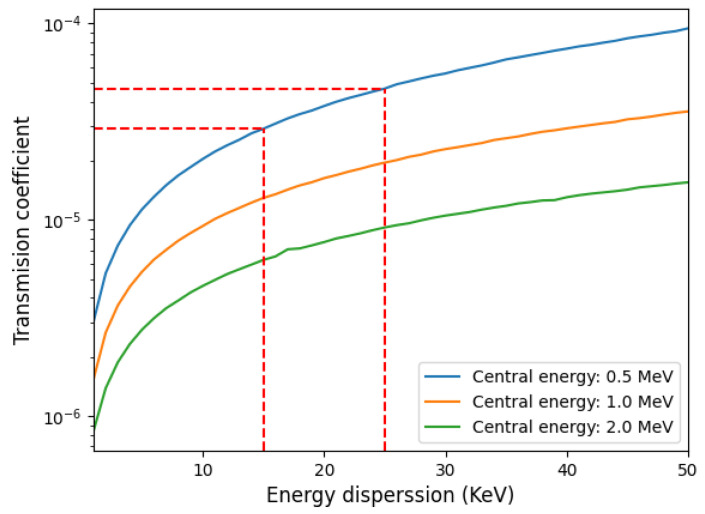
Transmission coefficient of the whole system as a function of energy spread for different central energies. Higher central energy results in a longer path through the dipole and a larger final spatial band at the exit, implying greater particle losses. Higher energy spread increases the particle flux.

**Figure 5 sensors-24-05254-f005:**
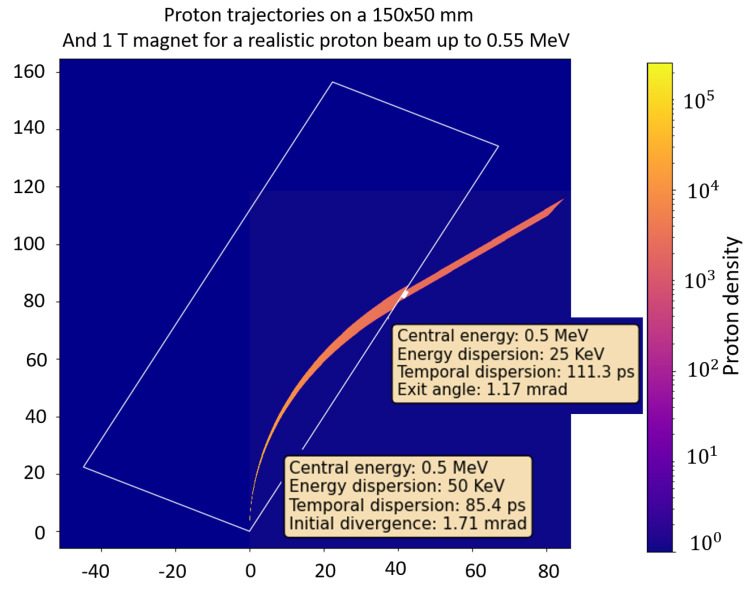
Example of particle tracking calculations with the following parameters: (i) distance between the source and the first slit of $d_{source}$ = 2 mm, (ii) distance between slits of 7.6 mm, (iii) slit width of 8 μm, (iv) dipole magnetic field of *B* = 1 T, (v) angle between the dipole side and the proton beam θ = 60^∘^ (30^∘^ from the vertical axis), (vi) dipole length 15 cm. The exit pinhole is placed $p_{pinh}$ = 10.17 cm from the entrance of the proton beam, and has a radius of 2.3 mm.

**Figure 6 sensors-24-05254-f006:**
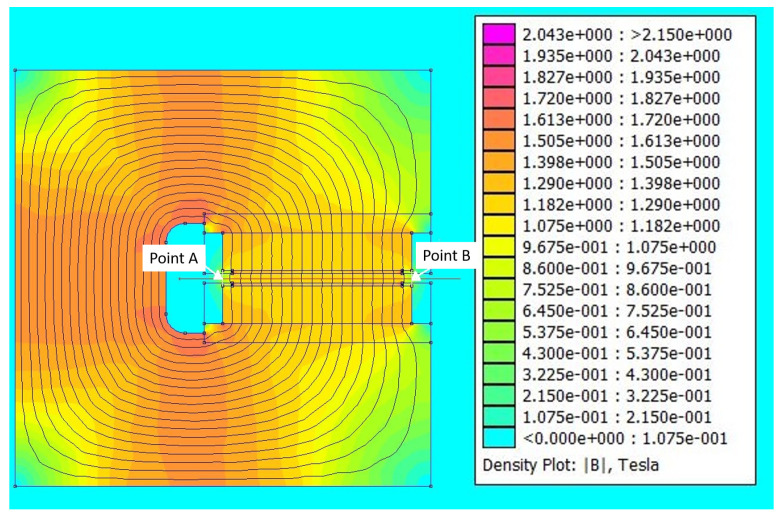
Simulation of the ferromagnetic structure to minimize border effects for a magnetic dipole composed of Pure Iron and Neodymium magnets (N52 grade, $4\times 10^{3}$ mm^3^), separated by 5 mm.

**Figure 7 sensors-24-05254-f007:**
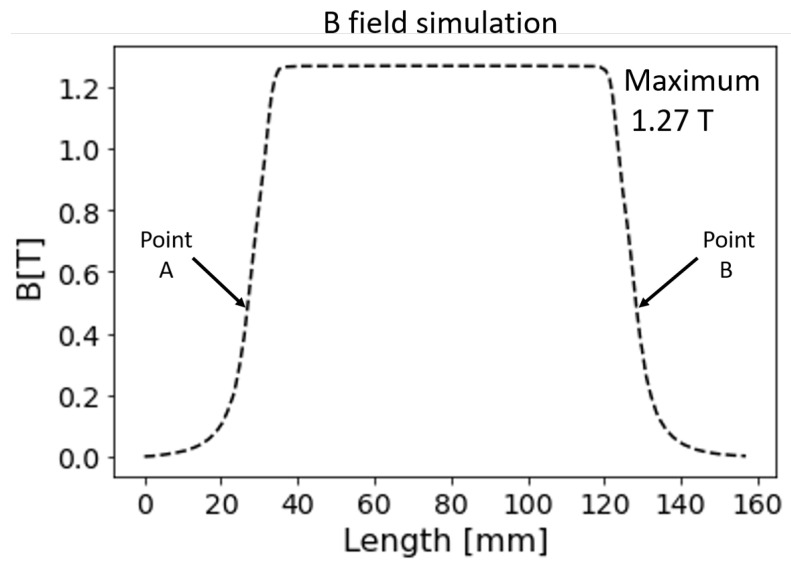
Line out of the simulated magnetic field in Figure 6 shows the field at Point A and Point B, where the magnet has a field of 0.5 T. The field outside the gap at 5 mm is 100 mT, and at 10 mm it is 50 mT. The field increases from 0.5 T to 1 T over approximately 6 mm and from 0.5 T to 1.25 T over 10 mm.

**Figure 8 sensors-24-05254-f008:**
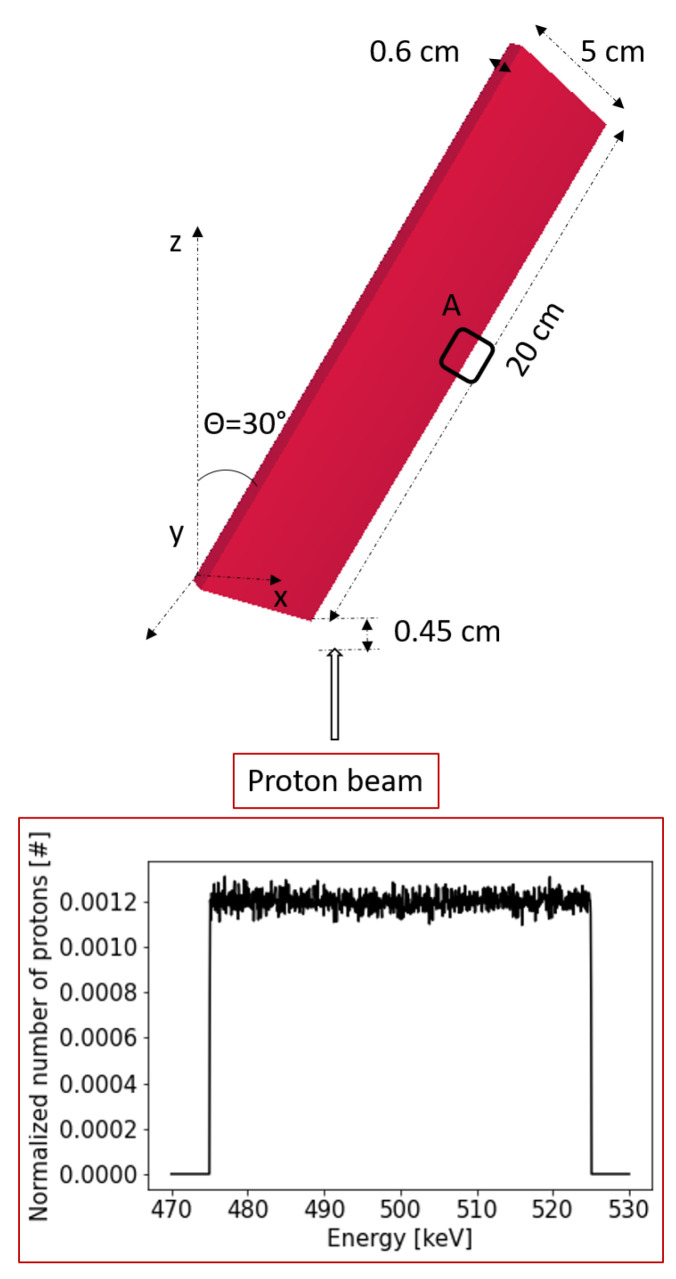
VisedX 3D view of the initial conditions for the MCNP6 simulation. The A zone is the output region of the proton beam, and the inset shows the proton energy distribution.

**Figure 9 sensors-24-05254-f009:**
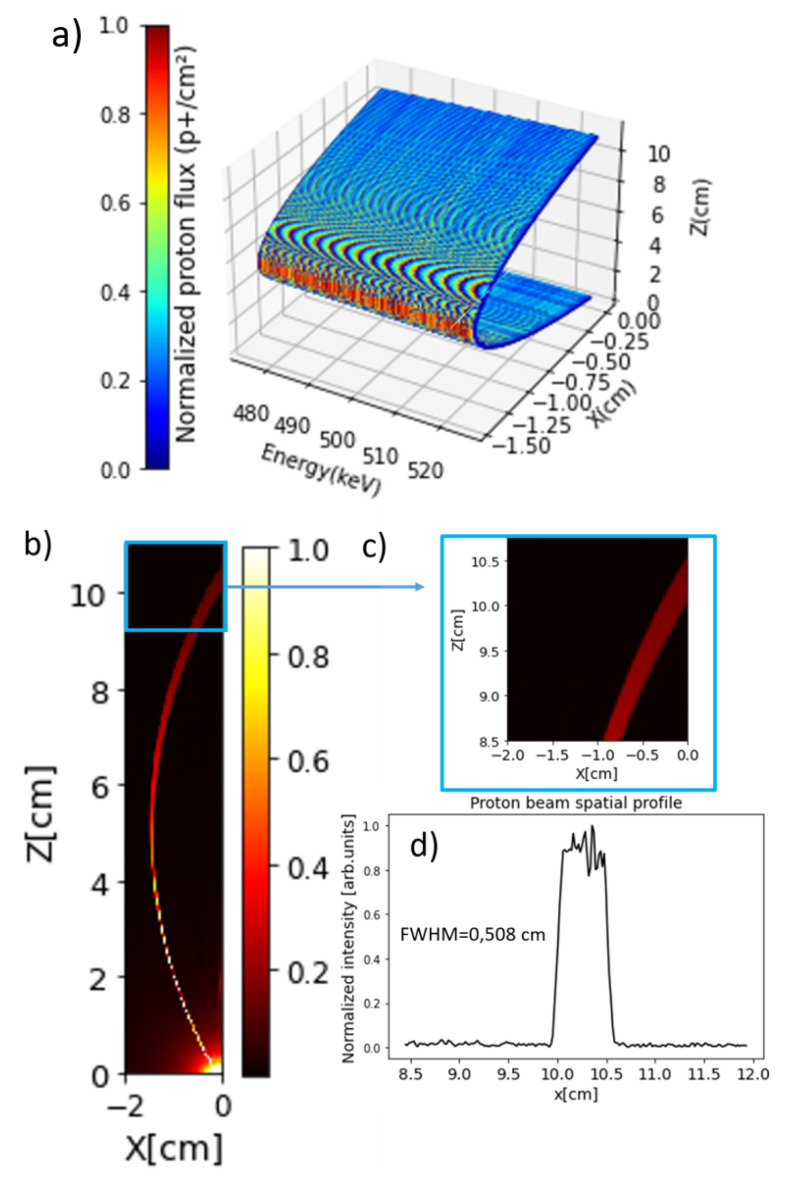
(**a**) MCNP6 proton trajectories as a function of energy. (**b**,**c**) Projection of the proton flux over the magnet surface with 0.2 degrees of divergence, and a detail of the magnet exit. (**d**) Normalized output proton profile at the magnet surface.

**Figure 10 sensors-24-05254-f010:**
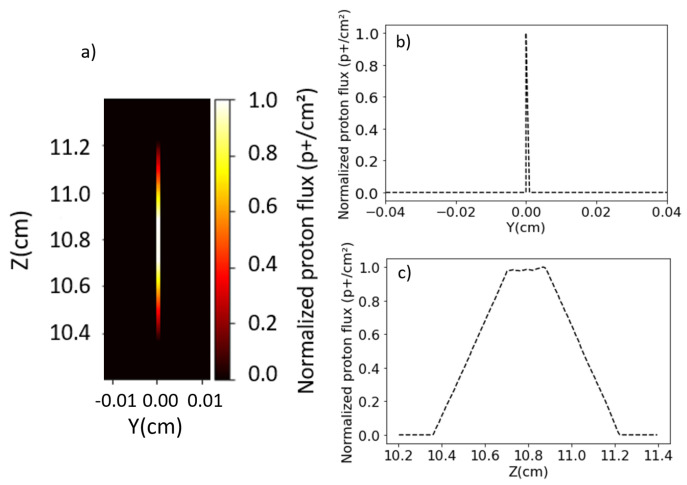
(**a**) Spot at the exit of the magnet. (**b**,**c**) are the vertical and horizontal profiles, respectively.

## Data Availability

Experimental data are stored by CLPU.

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
