# Peer review of "A Platform for Ultra-Fast Proton Probing of Matter in Extreme Conditions"

_sensors, 2024, doi:10.3390/s24165254_

Round 1

Reviewer 1 Report

Comments and Suggestions for Authors

The authors proposed in this manuscript 'A Platform for Ultra-fast Proton Probing of Matter in Extreme Conditions' which is well written and of high relevance to scientific community.  The authors claimed that the proposed device can finally select and measure proton beams with central energies of hundreds of keV and tens of keV bandwidth and may transport to the sample maintaining the temporal duration below 100s of ps. The manuscript needs more clarity in presentation and claim (as discussed in section 4.) before accepting for publication.

1. The english of the manuscript need to be enhanced,

2. Presentation quality of Figure (7) needs attention,

3. Section 4 is very brief summary of work shown in the manuscript and it may be extended to summarize the results. 

The authors also claim in this section that 'A preliminary parametric study is presented by comparing the performances with a previous magnetic selector used in a recent experiment.'. Its suggested to cite this parametric study in manuscript and which experiment is referred here. Its also suggested to refer the claimed numerical comparison and numerical simulation in last two lines of conclusion section.

Comments on the Quality of English Language

The english of the manuscript need to be enhanced,

Reviewer 2 Report

Comments and Suggestions for Authors

Reviewer 3 Report

Comments and Suggestions for Authors

The manuscript proposes an isochrone magnetic selector design for energy selection, and transport of laser-driven proton beams, which is claimed to be able to significantly reduce the temporal duration of the beams.

The basic idea is that the ions’ travel time in a bounded uniform magnetic field (with straight boundaries) depends only on the angle θ between the proton velocity direction and the magnetic field boundary. By optimizing the settings such as the distance between primary collimator slits, the position and size of the energy selector pinhole, the authors aim to minimize the beam's temporal dispersion. They presents a typical design where the output duration is less than 100 ps, with a maximum energy selection of 1.9 MeV. Analytical theory and numerical simulations validate the propose as potential applications for proton irradiation under high dose rate in various fields, including biomedical physics and materials science. I general, I don’t see serious scientific/technical problems in the MS except poor readability. But it apparently needs a big improvement before it’s qualified to be published. Below are some questions.

1.The introduction provides an extensive introduction to laser-driven proton sources and their applications in HEDP and biomedical fields. While this background is important, the introduction, all presented in a single paragraph, is too long and hard to read. A reform and polish is very necessary

2.In the current design, the duration shortening of the beam is realized by narrowing the incident angle and the exit slit. This sheme could be applied to any magnetic field as long as the trajectories of the ions are not crossed. What is the advantage of the proposed design if it does not collect ions with different incident angles to a certain point simultaneously but only selecting ions with specific energy and incident angle? Can the current design focus the beam at the exit slit?

3.In fig.1., the protons converge between the 2 slits. It’s unpractical to me. Besides, for fs-driving lasers and thin foil targets, the sources size is typically small, not 100s um as assumed. It’s more like a diverging point source instead of a converging source. Revision needs to be done in the discussions below based on a more practical source, .

4. In previous works by the authors (References 20 and 30), the energy selector of the magnetic selector was not close to the magnetic field boundary, leading to significant temporal dispersion in the output beam. Will the position of the energy selector in this design pose practical challenges?

5.The proposed isochrone magnetic selector has a very low transmission efficiency, resulting in a limited number of output protons. What is the expected flux for a typical 100s-TW acceleration result? What challenges will this bring to the application? How is does this selector behaves (efficiency, duration...)compared to a standard Thompson Spectrometer where the beam is collimated at the beginning as a pensil beam? A deeper discussion is needed.

6.How the non-uniformity of the magnetic field influences the temporal and energy dispersion of the output beam? Why don't use a magnet yoke to minimize the leakage of the magnetic field at the edge ?

7.I don’t really understand the fig 3. (b)(c)

8.I don’t really understand the fig 9. (a)

Minor problems:

n   "Developing ultra-short proton pulses was mainly driven from Ion stopping power in WDM experiments even if there are at least two more important applications of such pulses."- why use the words of “even if ” here?

n  line 108: what's MNP detector?

n  In fig.2, what's the assumed acceptance angles for the liens?

n  line 208: the word of phases is unsuitable here.

n  Formula (5) contains an error; some formulas lack numbering.

n  In line 29, the term "detector" is more appropriate than "sensor" for radiation diagnostics instruments.

n  In line 101, the magnetic separator used in references 20 and 30 were used after a magnetic spectrometer coupled with MCP.  

n  In Section 2.3, the authors did not calculate the beam transmission efficiency from the isochrone magnetic transporter through the energy selector (ηout).

n  the caption of fig. 3 is hard to understand, it needs a reform. MCNP6 is not explained before that.

n  There are many instances where there is no space between numbers and units. Consistent spacing should be ensured throughout the manuscript.

n  In line 276, it should be "0.5 MeV" instead of "0,5 MeV."

n  The analysis of the simulation results presented in Fig.5. is insurficient to me.

Comments on the Quality of English Language

Moderate editing of English language required
